# Gut Microbiota Markers and Dietary Habits Associated with Extreme Longevity in Healthy Sardinian Centenarians

**DOI:** 10.3390/nu14122436

**Published:** 2022-06-12

**Authors:** Vanessa Palmas, Silvia Pisanu, Veronica Madau, Emanuela Casula, Andrea Deledda, Roberto Cusano, Paolo Uva, Andrea Loviselli, Fernanda Velluzzi, Aldo Manzin

**Affiliations:** 1Department of Biomedical Sciences, Microbiology and Virology Unit, University of Cagliari, 09042 Monserrato, Italy; vanessapalmas@hotmail.it (V.P.); s.pisanu6@gmail.com (S.P.); veroma.dau91@gmail.com (V.M.); em.casula5@gmail.com (E.C.); 2Department of Medical Sciences and Public Health, University of Cagliari, 09124 Cagliari, Italy; andredele@tiscali.it (A.D.); alovise2@gmail.com (A.L.); fernandavelluzzi@gmail.com (F.V.); 3Interdisciplinary Center for Advanced Studies, Research and Development in Sardinia (CRS4), Science and Technology Park Polaris, Piscina Manna, 09134 Pula, Italy; robycuso@crs4.it; 4Clinical Bioinformatics Unit, IRCCS Istituto Giannina Gaslini, 16147 Genoa, Italy; paolouva@gaslini.org

**Keywords:** intestinal microbiota, gut dysbiosis, intestinal eubiosis, longevity, centenarians, 16S rRNA, inflammation, Mediterranean diet, lifestyle, bowel function

## Abstract

This study was aimed at characterizing the gut microbiota (GM) and its functional profile in two groups of Sardinian subjects with a long healthy life expectancy, overall named Long-Lived Subjects (LLS) [17 centenarians (CENT) and 29 nonagenarians (NON)] by comparing them to 46 healthy younger controls (CTLs). In addition, the contribution of genetics and environmental factors to the GM phenotype was assessed by comparing a subgroup of seven centenarian parents (CPAR) with a paired cohort of centenarians’ offspring (COFF). The analysis was performed through Next Generation Sequencing (NGS) of the V3 and V4 hypervariable region of the 16S rRNA gene on the MiSeq Illumina platform. The Verrucomicrobia phylum was identified as the main biomarker in CENT, together with its members *Verrucomicrobiaceae*, *Akkermansia* and *Akkermansia muciniphila*. In NON, the strongest associations concern Actinobacteria phylum, Bifidobacteriaceae and *Bifidobacterium*, while in CTLs were related to the Bacteroidetes phylum, Bacteroidaceae, *Bacteroides* and *Bacteroides* spp. Intestinal microbiota of CPAR and COFF did not differ significantly from each other. Significant correlations between bacterial taxa and clinical and lifestyle data, especially with Mediterranean diet adherence, were observed. We observed a harmonically balanced intestinal community structure in which the increase in taxa associated with intestinal health would limit and counteract the action of potentially pathogenic bacterial species in centenarians. The GM of long-lived individuals showed an intrinsic ability to adapt to changing environmental conditions, as confirmed by functional analysis. The GM analysis of centenarians’ offspring suggest that genetics and environmental factors act synergistically as a multifactorial cause in the modulation of GM towards a phenotype similar to that of centenarians, although these findings need to be confirmed by larger study cohorts and by prospective studies.

## 1. Introduction

The aging process is influenced and determined by complex interactions between genetic and environmental factors which, together with the stochastic process, can contribute to the attainment of longevity [1,2]. Evidence that relatives of people with a long lifespan are more likely to live longer and have a reduced risk of developing significant age-related diseases makes aging a potential therapeutic target [3,4,5].

Physiologically, aging is generally associated with low-grade systemic inflammation, determined by imbalances between pro-inflammatory and anti-inflammatory activity, which results in higher susceptibility to chronic diseases and disabilities [6,7,8]. At the gastrointestinal level, aging entails the impairment of intestinal motility and increased permeability, as well as changes to the intestinal nervous system and associated alterations in GM [9,10,11]. In addition, reduction of physical exercise, alteration of taste and smell, dyspepsia, dysphagia and complications related to teeth loss, are factors that exert a strong influence on the modification of the intestinal microbial community structure in aging [12,13,14,15,16].

Given its involvement in the aging process and its contribution to the regulation of immune response and metabolic homeostasis, the intestinal microbiota has been postulated as a possible biomarker of healthy aging [12,17,18]. In fact, the ability of eubiosis to contrast inflammation, alterations in intestinal permeability, as well as the decline of both cognitive and bone health, is well known [19]. When in its dysbiotic state, the GM is capable of affecting the homeostatic systems of the individual [20] and has been associated with diseases (and/or their progression), such as pro-inflammatory conditions [21], cardiovascular [22], neurological [23,24,25,26], hepatic [27], autoimmune diseases [28], as well as metabolic disorders, including obesity and metabolic syndrome [29,30,31] and inflammatory bowel diseases (IBD), such as ulcerative colitis and Crohn’s disease [32,33]. Moreover, GM composition has shown to be positively modulated by nutritional pattern, particularly by the Mediterranean Diet (MD), rich in fibres and antioxidant compounds [34,35].

An abundance of evidence links aging with GM alterations, e.g., decreased diversity, together with increased colonization by pathogenic microorganisms [19,36,37,38], thus several hypotheses have been made about the implications of the GM in the successful aging process. On one hand, it is believed that age-related GM remodeling may contribute to systemic inflammation, which in turn can directly or indirectly affect the gut community structure in a self-sustaining cycle [39]. On the other hand, it has also been hypothesized that the intestinal microbial rearrangements are intended to compensate for the harmful activities associated with the excessive presence of pathogens [19]. In this regard, Biagi et al. [14] describe the GM of long-lived people as a system able to maintain functional interaction with the host by successfully adapting to lifestyle changes and to potentially compromised physiological functions, which characterize advanced age.

To date, a specific microbial profile of successful aging has not been clearly defined, partially due to the variability of the inclusion factors under study and the different analytical methods used for analysis. In fact, factors such as geographic heterogeneity, residence structure and different dietary regimes can contribute to the heterogeneity of the results [9,18,37,40].

We focused the survey of our study on Sardinia, an Italian island, and more specifically to southern Sardinia. Sardinia represents an interesting geographical study target in this context, since it has been classified as one of the five “Blue Zones” among the five continents of the world where the highest instances of longevity and a higher frequency of long-lived males are found [41]. Furthermore, a high degree of internal heterogeneity has recently been highlighted in Sardinia; the subject of numerous studies for its genetic peculiarity due to its geographical isolation [42].

Previous studies have highlighted interesting intestinal microbial profiles [43] and bacterial and fungal communities in different anatomical sites [44] in long-lived subjects of Sardinian origin, who were, however, from northern Sardinia. We focused our investigation on southern Sardinia, thus expanding the literature on this topic. In particular, we analyzed intestinal microbiota and its metabolic function in healthy Sardinian centenarians compared to those of nonagenarian cohorts and those of a younger population. In addition, the contribution of genetics and environmental factors to the GM phenotype was assessed by comparing one subgroup of centenarians to a paired cohort of their children, in order to determine to what extent GM changes are involved in successful aging and whether age-related GM rearrangements are a contributing factor, or rather a consequence of, exceptional and healthy aging. Lastly, a correlation analysis between significant microbial taxa associated with longevity and dietary, lifestyle and clinical variables was performed.

## 2. Materials and Methods

### 2.1. Study Design and Characteristics of Subjects

The study evaluated the distinctive signatures of the GM and its functional profile in a sample of 92 subjects of Sardinian origin. Among them, two groups of subjects with a long and healthy life expectancy, referred to as LLS (17 CENT and 29 NON) were compared to 46 younger CTLs. In addition, the contribution of genetics to the GM phenotype was assessed by comparing a subgroup of seven CPAR with a paired cohort of the COFF.

Cases and controls, matched by gender, were recruited in the study by the Department of Biomedical Sciences and Department of Medical Sciences and Public Health of the University Hospital of Cagliari (Sardinia, Italy). As for inclusion criteria, the groups consisted of Sardinian subjects aged ≥ 100 and ≥ 90 for CENT and NON groups, respectively, and aged between 40 and 60 for CTLs. Healthy subjects with a suitable index of independence in daily life activities and of cognitive ability (assessed by “Activities of Daily Living test” and “Mini Mental State Examination test”, respectively) were also required. The exclusion criteria included the inability to provide written informed consent or to follow the procedures determined by the protocol, diagnosis of malignant neoplasm and/or history of malignant neoplasms surgically removed within five years prior to enrollment, IBD, celiac disease, uncontrolled diabetes, serious cardiovascular disease, cardiovascular events within five years prior to enrollment, and other clinically significant severe pathologies such as renal, hepatic, hematological, pulmonary, neurological, psychiatric, immunological, gastrointestinal or endocrine diseases, therapy with immunosuppressive drugs (cyclosporine, methotrexate, glucocorticoids) or anticoagulants, antimicrobials and/or prebiotic or probiotic intake, and any hypocaloric diet in the 60 days preceding the sample collection.

Clinical data from each study participant, including demographic and anthropometric data, lifestyle factors (smoking status, alcohol and coffee consumption, bowel function, medications and Mediterranean Diet score (MDS)) and the presence of comorbidities, were collected (Table 1). Limited to the LLS group, data relating to Mini Mental State Evaluation (MMSE), Activities of Daily Living (ADL), Mini Nutritional Assessment (MNA) and Physical Activity Scale for the Elderly (PASE) were also recorded. All clinical evaluations were performed contextually to the sampling date. The anthropometric and lifestyle factor assessments, such as MMSE [45], ADL [46], MNA [47,48], PASE [49] and MDS [50] scores can be found as Appendix A in the sections “Anthropometric evaluation” and “Lifestyle factor assessment”.

### 2.2. Sampling

Stool samples from each subject were independently collected. The collection was carried out at home or in the host structures by the staff, using standard safety procedures. Transport was carried out by the staff and delivery was made to the laboratory within 3 h. Fresh samples were stored at −80 °C until further processing.

### 2.3. Total DNA Extraction from Fecal Sample and Quantification of Bacterial DNA

Genomic DNA was isolated from the fecal sample utilizing the QIAamp Fast DNA Stool Mini Kit following the manufacturer’s instructions (Qiagen, Hilden, Germany). The concentration of the fecal bacterial DNA of each patient was quantified through real-time PCR (qPCR) on the genomic DNA samples, performed using degenerate primers encompassing the V3 and V4 hypervariable region of the bacterial 16S rRNA gene, as previously described [32].

### 2.4. 16S Libraries Preparation and Sequencing

The protocol of library preparation and sequencing has been described in detail elsewhere [29]. 16S barcoded amplicon libraries were generated using primers targeting the V3-V4 hypervariable region of the bacterial 16S rRNA gene and the Nextera XT index kit (Illumina, inc., San Diego, CA, USA), and their size and quality were verified using Agilent DNA 1000 Analysis kit (Agilent Technologies, Santa Clara, CA, USA) on the Agilent 2100 Bioanalyzer system (Model G2939B, Agilent Technologies, Santa Clara, CA, USA). Genomic libraries were quantified with a Qubit 3.0 Fluorometer (Thermo Fisher Scientific, Waltham, MA, USA) using the Qubit dsDNA HS Assay Kit (Thermo Fisher Scientific, Waltham, MA, USA), normalized to a concentration equal to 4 nM, then pooled. The pooled library, and the adapter-ligated library PhiX v3 used as a control, were denatured and diluted to equal concentration (8 pM) and subsequently combined to obtain a PhiX concentration equal to 5% of the total volume. Combined 16S library and PhiX control were further denatured and sequenced on the MiSeq platform using MiSeq v3 Reagent Kit (Illumina).

### 2.5. Data and Statistical Analysis

Analysis of the data generated on the Miseq System was carried out using the BaseSpace 16S Metagenomics App (Illumina), whereas operational taxonomic unit (OTU) mapping to the Greengenes database (v.13.8) [51] was performed using the Quantitative Insights Into Microbial Ecology (QIIME) platform (v.1.8.0) [52].

Alpha diversity was assessed with the script alpha rarefaction.py in QIIME in order to obtain the Shannon index. Alpha diversity and Firmicutes/Bacteroidetes ratio were analyzed using the Kruskal-Wallis test followed by Bonferroni correction for multiple comparisons. Beta diversity was generated in R-vegan, using the Bray-Curtis distance.

The Non-Metric Multidimensional Scaling (NMDS) based on the Bray-Curtis distance matrix was conducted in R software v.3.5.2 (ggplot2 package). The statistical significance of beta diversity among the groups was determined with Permutational Multivariate Analysis of Variance (PERMANOVA) (R-vegan, function adonis). The overall *p*-value obtained from multiple comparisons was confirmed through a pairwise PERMANOVA test performed in R (RVAdeMemoire package). The analysis at taxonomic levels was performed in SPSS software v.28.0.1.0 (IBM, New York, NY, USA) using the Kruskal-Wallis test. Pairwise comparison was performed only for significant taxa, followed by Bonferroni correction for multiple comparisons. Only bacteria present in at least 25% of the samples and with a relative abundance of ≥0.1% in cases and/or controls were considered. The Linear Discriminant Analysis Effect Size (LEfSe) was additionally performed on statistically significant bacterial taxa obtained by the Kruskal-Wallis test and confirmed after Bonferroni adjustment. The LEfSe algorithm was performed on the Galaxy computational tool v.1.0. (http://huttenhower.sph.harvard.edu/galaxy/) accessed on 30 August 2021. The association between the relative abundance of significant taxonomic levels and dietary, lifestyle, and clinical variables was evaluated by calculating the Spearman’s correlation on SPSS software v.28.0.1.0.

Phylogenetic Investigation of Communities by Reconstruction of Unobserved States (PICRUSt) [53] algorithm was performed on Galaxy software v.1.0. (https://galaxy.morganlangille.com/, accessed on 30 August 2021) to infer metagenome composition in the samples by analyzing OTUs generated by QIIME pipeline. Bacterial metabolic pathways were predicted and classified by Kyoto Encyclopedia of Genes and Genomes (KEGG) [54].

Statistical differences were analyzed for all metabolism pathways present in at least 25% of the samples and with a minimum abundance of 0.1% using Statistical Analysis of Metagenomic Profiles (STAMP) software [55]. The statistical significance was tested using Welch’s test, with a Storey False Discovery Rate correction (FDR) correction. Overall, *p* ≤ 0.05 was considered statistically significant.

## 3. Results

### 3.1. Clinical and Lifestyle Data of Subjects

Clinical characteristics of the study cohorts are shown in Table 1 and Table 2. Overall, the study cohorts were rather homogeneous with each other regarding demographic, anthropometric and lifestyle data. Notably, the analysis of clinical data showed no statistically significant differences between CENT and NON in terms of gender, BMI, comorbidities and lifestyle factors, except for MMSE, ADL and PASE scores, while LLS groups diverged significantly from CTLs in some clinical factors, such as BMI, number of medications per day and comorbidities. Furthermore, the number of smokers was significantly higher in CTLs than in the NON group (*p* = 0.027) and NON presented a lower adherence to MD compared to CTLs, though the statistical value was not highly significant (*p* = 0.036). The CPAR and COFF cohorts were not significantly different in terms of gender, BMI and most lifestyle factors, except bowel function, number of medications, MMSE, ADL, PASE and MNA scores.

### 3.2. Gut Microbiota Analysis

#### 3.2.1. Alpha and Beta Diversity Analysis

The Kruskal-Wallis test showed statistically significant differences in the Shannon index across different study cohorts (*p* = 0.037), confirmed by pairwise testing only for the long-lived group comparison that showed an alpha diversity in CENT significantly lower than that in NON (CENT = 2.39 ± 0.31, NON = 2.46 ± 0.29, *p* = 0.020). Alpha diversity was higher in both LLS cohorts compared to controls, albeit not significantly (see Figure 1A and Appendix A online). No statistically significant differences in the Shannon index between CPAR and COFF subgroups were observed (*p* = 0.398; see Appendix A online).

The Non-Metric Multidimensional Scaling (NMDS) based on the Bray-Curtis distance matrix showed a marked separation between the GM communities of LLS and CLTs (see Figure 1B and Appendix A online), confirmed by PERMANOVA analysis, which indicated a significant difference in beta diversity between cohorts (sum of squares = 1.498, mean of squares = 0.749, F = 6.074, R = 0.1201, *p* = 0.001). Significant segregation persisted only in the comparison between CENT and CTLs (*p* = 0.006) and between NON and CTLs (*p* = 0.003) following the pairwise PERMANOVA test.

No statistically significant differences in beta diversity between CPAR and COFF subgroups were obtained (sum of squares = 0.08, mean of squares = 0.08, F = 0.706, R = 0.056, *p* = 1) (Appendix A online).

#### 3.2.2. Compositional Analysis of the Gut Microbiota

Illumina MiSeq generated a mean value of 108,775 (+/− 16,901 SD) reads per patient.

The Firmicutes/Bacteroidetes ratio was significantly higher in LLS compared to CTLs (see Table 3), while no statistical significance persisted when CPAR and COFF were compared (*p* = 0.499).

The Kruskal-Wallis test on GM composition between CENT, NON and CTLs showed 105 statistically significant results (see Table 4). Pairwise analysis showed 29 common significant differences in the two classes of LLS compared to CTLs, 24 significant divergences were found only from the comparison between CENT and CTLs, 41 only from the comparison between NON and CTLs, while the GM of CENT and NON differed significantly in 8 bacterial taxa (see Appendix A online).

The Kruskal-Wallis test on GM composition between CPAR and COFF showed five statistically significant results, which did not maintain statistical significance after Bonferroni correction (see Table 5).

The LEfSe was additionally performed on statistically significant bacterial taxa obtained by the Kruskal- Wallis test and confirmed after Bonferroni adjustment. Results were ranked by their Linear Discriminant Analysis (LDA) score (see Figure 2): the Verrucomicrobia phylum was identified as the main biomarker in CENT, together with its members *Verrucomicrobiaceae*, *Akkermansia*, *Akkermansia muciniphila*, *Prosthecobacter*, Luteolibacter, *Luteolibacter algae*, *Rubritaleaceae* and *Rubritalea*, while within the Firmicutes phylum the strongest associations were related to *Acidaminococcaceae*, *Phascolarctobacterium* and *Phascolarctobacterium faecium*. Strongly associated were the Synergistetes phylum and related Synergistaceae and *Cloacibacillus* taxa, as well as *Xanthomonadaceae*, *Desulfonauticus*, *Desulfonauticus autotrophicus* (Proteobacteria) and *Eggerthella*, *Bifidobacterium bifidum* and *Collinsella intestinalis* (Actinobacteria). Euryarchaeota phylum and related taxa Methanobacteriaceae, *Methanobrevibacter*, *Methanobrevibacter smithii* also showed a strong association in CENT, as well as Rikenellaceae, *Bacteroides intestinalis*, *Prevotella* and *Prevotella shahii,* relative to Bacteroidetes phylum.

In CTLs, the strongest associations were related to Bacteroidetes phylum and its members Bacteroidaceae, *Bacteroides* and *Bacteroides* spp. In the same cohort, the main biomarkers were taxa belonging to Cyanobacteria phylum, such as Aphanizomenonaceae, *Dolichospermum* and *Dolichospermum macrosporum*, to Proteobacteria phylum (Alcaligenaceae, Comamonadaceae*,* Oxalobacteraceae, Collimonas, *Candidatus Blochmannia*, *Candidatus Blochmannia rufipes*, Sutterellaceae and *Sutterella*), and to Firmicutes phylum, such as *Acidaminococcus intestini*, Erysipelothricaceae, *Erysipelothrix*, *Erysipelothrix inopinata*, *Butyrivibrio* and *Butyrivibrio proteoclasticus*. The Prevotellaceae family (Bacteroidetes phylum), also showed strong association in CTLs.

In NON, the strongest associations concern Actinobacteria phylum and related members Bifidobacteriaceae, *Bifidobacterium* spp., Coriobacteriaceae, *Collinsella*, *Collinsella tanakaei*, Eggerthellaceae, *Slackia* and *Blautia coccoides*. A strong association was also found for the taxa Streptococcaceae, *Streptococcus*, *Streptococcus* spp., Lactobacillaceae, *Lactobacillus*, *Lactobacillus* spp., Veillonellaceae, *Veillonella*, *Veillonella dispar*, *Veillonella atypica*, Thermicanaceae, *Thermicanus*, Bacillales_Family X_Incertae Sedis, all belonging to Firmicutes phylum. In the Bacteroidetes phylum an association was found only for the *Bacteroides fragilis* species. Desulfovibrionaceae, *Desulfovibrio* and *Desulfovibrio piger* (Proteobacteria) were also strongly associated in NON.

LEfSe plots of taxonomic biomarkers were generated on the Galaxy computational tool v.1.0. (https://huttenhower.sph.harvard.edu/galaxy/) accessed on 30 August 2021. Results were ranked by their Linear Discriminant Analysis (LDA) score. Blue bacterial taxa were more abundant in NON, green bacterial taxa were more abundant in CTLs, red bacterial taxa were more abundant in CENT. CENT = centenarian subjects, NON = nonagenarian subjects, CTLs = healthy younger controls.

#### 3.2.3. Spearman Correlation between Gut Microbiota Alterations and Dietary, Lifestyle and Clinical Variables in CENT and Non

Taxa significantly associated with longevity were correlated with dietary, lifestyle and clinical parameters in both CENT and NON. Most taxa associated with CENT were correlated with MDS score and bowel function (see Figure 3A and Appendix A online). As for the former, seven bacterial taxa were positively correlated, while six were negatively correlated. The taxa related to MDS score mainly concerned the Firmicutes phylum, such as *Lactobacillus taiwanensis*, Clostridiaceae and its members Clostridium and *Dorea*, *Peptoniphilus*, Thermicanaceae and *Thermicanus*, all of which were positively correlated, and *Catenibacterium*, *Veillonella* and *Dialister invisus*, which were all negatively correlated. Furthermore, *Bacteroides rodentium* and *Parabacteroides merdae*, which belong to Bacteroidetes phylum, and *Eggerthella*, which belongs to Actinobacteria phylum, were negatively and positively correlated with MDS, respectively. As for bowel function, eight bacterial taxa were positively correlated, and one was negatively correlated. The positively correlated taxa mostly concerned the Firmicutes phylum (Thermicanaceae, *Thermicanus*, *E. inopinata*), but also included Desulfohalobiaceae, *Desulfonauticus*, *D. autotrophicus* (Proteobacteria) and Synergistetes phylum with its Synergistaceae family. The taxa negatively correlated to bowel function belonged to the Firmicutes phylum (*L. taiwanensis*). Some bacterial taxa have frequently shown a correlation with several clinical variables. The Thermicanaceae family and related *Thermicanus* genus were also positively related to levels of current and former alcohol consumption, whereas *Desulfovibrio* was negatively correlated with MMSE and MNA scores and to former alcohol consumption; *D. piger* negatively correlated with ADL, MNA and PASE scores.

In NON, there was a greater number of correlations than in CENT (see Figure 3B and Appendix A online), mainly related to the number of medications, MDS and PASE score. Taxa belonging to Actinobacteria phylum, such as *Bifidobacterium*, *Bifidobacterium* spp., *Blautia wexlerae* (Firmicutes) and Bacteroidaceae (Bacteroidetes phylum) were negatively correlated to the number of medications, while the Bacteroidetes phylum (and related *Bacteroides*, *Bacteroides* spp., *Sphingobacterium shayense*) and Alcaligenaceae and *Sutterella* (Proteobacteria phylum) showed a positive correlation. Members belonging to the Actinobacteria phylum were all negatively correlated to MDS scores (Bifidobacteriaceae, *Bifidobacterium*, *Bifidobacterium* spp., Streptomycetaceae), as were those belonging to Bacteroidetes (Odoribacteraceae, *Bacteroides* spp., *Pedobacter kwangyangensis* and *Parabacteroides* spp.) and to Firmicutes (*Alkaliphilus*, *Clostridium frigoris*, *Lactobacillus ultunensis*, *Peptoniphilus*). Several significant associations were found for other clinical variables, such as ADL and PASE scores. Taxa belonging to Proteobacteria phylum (*Serratia*, *S. entomophila*, *Escherichia*, *E. albertii*) and *Eggerthella* (Actinobacteria) were inversely related to ADL scores. In relation to PASE scores, Firmicutes members (Thermicanaceae, *Thermicanus*, *Blautia wexlerae*), on one hand, and Bacteroidetes and Proteobacteria members, on the other, showed a positive and negative correlation, respectively.

#### 3.2.4. Functional Metagenome Prediction Analysis

A comparative prediction analysis of the functional metagenome was performed using PICRUSt. A total of three different significantly metabolic pathways were identified by comparing CENT and NON (see Figure 4A). In particular, the pathway related to the biosynthesis of secondary metabolites (tropane, piperidine and pyridine alkaloid) were most expressed in CENT, while the pathways related to lipid metabolism (Ether lipid metabolism) and amino acid metabolism (D-Arginine and D-ornithine metabolism) were enriched in NON. The comparative functional metagenome prediction between LLS groups and CTLs showed a common significative decrease in glycan degradation in both LLS groups and a significant increase in secretion systems and in signal transduction (two-component system) in the same subjects (Figure 4B,C). In CENT the bacterial secretion system and the pathway of replication, recombination and reparation of proteins were also most expressed compared to CTLs; on the other hand, the metabolism of pyrimidine, amino and nucleotide sugar was reduced.

Comparing the functional metagenome prediction profile of NON and CTLs has shown that in the former, a significant decrease in starch and sucrose metabolism was observed, whereas in the latter, a reduction in transporters and ABC transporter pathways was found. No statistically significant differences in the functional metagenome comparing CPAR and COFF subgroups were observed.

## 4. Discussion

The present study aimed at characterizing human GM and its functional profile in two groups of Sardinian subjects with long, healthy lifespans (17 CENT and 29 NON) by comparing them to 46 younger CTLs. In addition, the contribution of genetics and environmental factors to the GM phenotype was assessed by comparing a subgroup of centenarian parents (CPAR) with a paired cohort of centenarians’ offspring (COFF). The analysis was performed through NGS of the V3 and V4 hypervariable regions of the 16S bacterial rRNA gene on the MiSeq Illumina platform.

The alpha diversity in CENT and NON was higher than in CTLs, although no significant difference in the Shannon index was observed. These data confirm previous studies’ results, as the literature agrees on a greater alpha diversity being associated with aging among elderly and extremely elderly adults; however, findings are not always statistically significant [56,57].

Beta diversity analysis showed a significant dissimilarity between both groups of long-lived subjects (CENT and NON) compared to CTLs, as previously reported [19,43,58,59].

Our microbial diversity findings represent a strong indicator of GM implications in advanced aging, in agreement with previous hypotheses. In fact, greater alpha diversity in both CENT and NON cohorts compared to healthy controls reflects a rich and complex microbial ecosystem, indicative of an adaptable intestinal microbiota, capable of adapting to multiple environmental perturbations. Due to this peculiarity, high intestinal microbial diversity has been defined as an indicator of longevity [60].

We also observed a statistically significant reduction in the alpha diversity of CENT compared to NON. It should be pointed out that lower alpha diversity has been associated with poor cognitive function [61], in line with a significantly lower MMSE score in the CENT cohort compared to NON. Furthermore, although the CENT cohort represents a healthy population, during aging, and especially in advanced aging, a physiological reduction in gastrointestinal function and host immune response has been observed linked to the development of chronic low-grade inflammation [6]. In this regard, it has been observed that reduced alpha diversity is related to metabolic and inflammatory diseases [62,63]. Therefore, a reduction in alpha diversity in CENT compared to NON is not surprising.

At the taxonomic level, we observed a significant increase in the Firmicutes/Bacteroidetes ratio in both CENT and NON groups compared to CTLs, in contrast to previous studies on centenarian subjects with the same [43] or a different geographic origin [36] and also to studies on the elderly [9,64]. Notably, among our CENT, NON and CTLs cohorts, the relative abundance of Firmicutes was approximately the same and did not change significantly (40.54, 47.72 and 43.15, respectively; *p* = 0.387), while that of Bacteroidetes almost doubled in CTLs compared to CENT and NON groups (43.59, 24.74 and 25.28, respectively; *p*_CENT*vs*CTLs_ = 0.015 *p*_NON*vs*CTLs_ = 0.022). In other words, the higher Firmicutes/Bacteroidetes ratio in our long-living subjects compared to controls reflected a significant reduction in the relative abundance of the Bacteroidetes phylum, rather than an increase in that of Firmicutes.

Multilevel taxonomic analysis showed greater divergences between LLS and CTLs, while the GM composition between CENT and NON did not diverge considerably. Specifically, more statistically significant differences (n = 41 taxa) were found between NON and CTLs. Twenty-four taxa significantly diverged when comparing CENT to CTLs, while twenty-nine common taxa in both CENT and NON were significantly altered compared to CTLs. Lastly, only eight bacterial taxa were significantly altered when comparing CENT to NON. This demonstrates that the GMs of the older cohorts are more similar to each other than to the cohort of younger subjects.

LEfSe showed that the Verrucomicrobia phylum was identified as the main biomarker in CENT, together with its members Verrucomicrobiaceae, *Akkermansia* and *Akkermansia muciniphila*, as often reported in previous studies of centenarians [57,65,66], while one study observed an opposite trend [64]. In addition, *Akkermansia* has been reported to increase with aging in several studies [6]. *Akkermansia muciniphila* is a mucin degrading bacterium that resides in the human intestinal mucous layer and is able of promoting intestinal integrity due to its capacity for restoring mucous thickness and thus stimulating the mucous turnover rate [67,68]. It is considered a significant biomarker of intestinal homeostasis, as its depletion has been associated with many diseases such as inflammatory bowel diseases and metabolic disorders [69]. Several studies confirm its protective effects. *A. muciniphila* has been reported to increase anti-tumor responses during anti-programmed cell death protein 1 (PD-1) immunotherapy [70], improve metabolic *status* and clinical outcomes after a dietary intervention in overweight/obese adults [71] and have protective effects in diet-induced obesity [72,73]. *A. muciniphila* supplementation in patients with overweight/obesity has reduced inflammation marker levels and improved several metabolic parameters [74], while in animal models of diabetes and obesity, it has restored the integrity of the epithelial mucosa, improved glucose tolerance and improved metabolic parameters, such as endotoxemia and inflammation [75].

Other strongly associated taxa in CENT were the Synergistaceae family, which belong to the Synergistetes phylum, *Eggerthella, Collinsella intestinalis* and *Bifidobacterium bifidum* (Actinobacteria), *Methanobrevibacter* and *Methanobrevibacter smithii* (Euryarchaeota phylum), as well as Rikenellaceae and *Prevotella* within the Bacteroidetes phylum. These associations are consistent with previous studies in which an increase in the abundance of all these taxa, with the exception of Prevotella, was observed in centenarians compared to younger subjects [19,43,58,66].

Some strongly associated taxa in CENT have never been reported before: *Prosthecobacter*, Luteolibacter, *Luteolibacter algae*, Rubritaleaceae and *Rubritalea*, belonging to the Verrucomicrobia phylum; Acidaminococcaceae, *Phascolarctobacterium* and *Phascolarctobacterium faecium* (Firmicutes phylum), the Synergistetes phylum and related *Cloacibacillus* taxa, Xanthomonadaceae, *Desulfonauticus* and *Desulfonauticus autotrophicus* (Proteobacteria), the Euryarchaeota phylum and related taxa Methanobacteriaceae, in addition to *Bacteroides intestinalis* and *Prevotella shahii*, relative of the Bacteroidetes phylum.

To date, the metabolic role of most of the taxa mentioned above has not been characterized. However, the beneficial effect of some of them has been described. For instance, *Bifidobacterium bifidum* constitutes one of the most dominant taxa of human intestinal microbiota in healthy breast-fed infants [76] and it has been individuated as one of the most abundant Bifidobacteria species in Italian centenarians [77].

*Bacteroides intestinalis* is able to degrade complex arabinoxylans from dietary fibre with the consequent release of the beneficial ferulic acid metabolite. It has been demonstrated that cultured *Bacteroides intestinalis* in the presence of complex insoluble arabinoxylans enhances the Th1-type immune response in mice and exerts anti-inflammatory activity in dendritic cells under inflammatory conditions [78].

*Phascolarctobacterium* is an acetate/propionate-producer, whose increase, observed after treatment with berberine and metformin in high-fat diet-induced obesity in rats, has been hypothesized to contribute to the beneficial effects of these two drugs [79]. *Phascolarctobacterium faecium* exerted beneficial effects on the host in rat models with nonalcoholic fatty liver disease [80], and has been associated with the supplementation of cruciferous vegetables in a controlled fruit and vegetable-free diet [81].

Regarding the Euryarchaeota phylum, the Methanobacteriaceae family and its members, *Methanobrevibacter* and *Methanobrevibacter smithii,* were strongly associated with CENT. These data are in agreement with previous studies that reported a high abundance of *Methanobrevibacter smithii* and of *Methanobrevibacter* genus in the centenarian gut microbiota of Sardinian and Chinese subjects, respectively [43,82]. *Methanobrevibacter smithii* represents the most dominant methanogen in the human gut due to its ability to reduce CO_2_ by using H_2_ (or formate) [83]. Several studies have investigated the possible link between the presence of methanogens and some human diseases, such as colorectal cancers (CRC), inflammatory bowel disease (IBD), irritable bowel syndrome (IBS), obesity and constipation, although contradictory findings make them difficult to interpret [83]. However, the mutualistic activity of *M. smithii* and *B. thetaiotaomicron*, inoculated into a germ-free mice model, has shown to promote an increase in caloric intake from diets, in lipogenesis and in host fat [84]. Noteworthy is the relationship between methanogens and aging, based on the excreting methane breath test, which has been consistently observed [83]. This association has not yet been defined, but different hypotheses have been formulated. Among them, the possibility that methanogens are selected during aging for their insensitivity to most of the antibiotics used throughout a lifespan, their slower transit time observed in aging, which possibly contributes to their over-representation, and favorable environmental exposure to methanogens in extreme long-living subjects during their life compared to current adults (different dietary habits, exposure to livestock) [83].

In NON, the strongest associations observed concerned the Actinobacteria phylum and related members Bifidobacteriaceae, Bifidobacterium angulatum, Bifidobacterium asteroides, Bifidobacterium catenulatum, Bifidobacterium choerinum, Bifidobacterium indicum and Bifidobacterium kashiwanohense. It should be pointed out that the abundance of these taxa was elevated in NON compared with CTLs and underwent a slight decrease in extreme longevity (CENT) compared with NON, albeit still higher than in CTLs. Interestingly, several studies have demonstrated that Bifidobacteria and its species diversity are decreased in elderlies [85]. Consistent with our findings, higher proportions have been identified in successful aging compared to younger elderlies or younger adult [14,43,86,87]. A reduction in Bifidobacteria has been associated with impaired adhesion to the intestinal mucosa, but it remains to be clarified whether the cause is attributable to changes in the mucus structure in the microbiota of elderly subjects [85]. Furthermore, Bifidobacteria depletion has been correlated with enhanced susceptibility to infections and impaired intestinal activity [88].

A strong association in NON was also found for Lactobacillaceae, *Lactobacillus* and *Lactobacillus* spp., which all belong to the Firmicutes phylum.

The probiotic effect of Lactobacillus and Bifidobacterium taxa has been well documented [89]. It has been demonstrated that *Lactobacillus* spp. or *Bifidobacterium* spp. probiotic supplementation attenuates oxidative stress and inflammation and improves physiological parameters such as gut barrier function, learning and memory ability in aged mice [90]. In addition, improved immunity in elderly humans and aged mice has been observed [91].

The taxa Streptococcaceae, *Streptococcus*, *Streptococcus* spp., Veillonellaceae, *Veillonella*, *Veillonella dispar*, *Veillonella atypica*, Thermicanaceae, *Thermicanus*, Bacillales_Family X_Incertae Sedis (Firmicutes phylum), Desulfovibrionaceae, *Desulfovibrio*, *Desulfovibrio piger* (Proteobacteria) and *Bacteroides fragilis* (Bacteroidetes) were also strongly associated with NON. *Veillonella* species are known to use lactic acid as a source of carbon and energy and are believed to ferment the lactic acid produced by Streptococcus, derived from the fermentation of carbohydrates [92], and positively correlates with the abundance of *Streptococcus* in irritable bowel syndrome (IBS) [93,94]. Furthermore, this bacterial genus has the ability to ferment organic acids through the production of propionic and acetic acids, carbon dioxide and hydrogen. The propionic acid produced by *Veillonella* potentially presents greater risks than benefits regarding the neurotoxic character linked to its accumulation [95,96].

As for *Desulfovibrio* bacteria, it has been demonstrated, using a Stress-Induced Premature Senescence Model of Bmi-1 Deficiency, that mice in whom this bacterial genus penetrated the epithelium underwent an induced TNF-α secretion by macrophages, which caused impairment of TNF-α-dependent intestinal barrier permeability and aging. Furthermore, *Desulfovibrio*, one of the predominant sulphate-reducing bacterial generates residing in the human gut, is capable of leading to the formation of hydrogen sulfide, which is toxic for intestinal epithelial cells and exerts a pro-inflammatory effect. In fact, its abundance has been correlated with IBD [97,98] and obesity [29].

In CTLs, the strongest associations were related to the Bacteroidetes phylum and its members Bacteroidaceae, *Bacteroides*, *Bacteroides* spp. and the Prevotellaceae family. In the same cohort, the main biomarkers were taxa belonging to the Cyanobacteria phylum, such as Aphanizomenonaceae, *Dolichospermum* and *Dolichospermum macrosporum*, to Proteobacteria phylum (Alcaligenaceae, Comamonadaceae*,* Oxalobacteraceae, *Collimonas*, *Candidatus Blochmannia*, *Candidatus Blochmannia rufipes*, Sutterellaceae and *Sutterella*) and to the Firmicutes phylum, such as *Acidaminococcus intestini*, Erysipelothricaceae, *Erysipelothrix*, *Erysipelothrix inopinata*, *Butyrivibrio* and *Butyrivibrio proteoclasticus*.

The association with the Bacteroidetes phylum was in line with previous studies [19,77,99] and disagrees with the initial hypothesis concerning the increase in the abundance of Bacteroidetes in old age, and with the reduction of the Firmicutes/Bacteroidetes ratio in older adults [6,64]. It should be borne in mind that the harmonic balance between Firmicutes and Bacteroidetes phyla in the human microbiota can be indicative of good health, but it is subject to the influence of lifestyle factors. The significant reduction in Bacteroidetes in both CENT and NON subjects compared to CTLs could be explained by the significantly higher BMI in these cohorts than in controls, given that Bacteroidetes are known to positively correlate with a reduction in body fat [29]. Furthermore, the relative abundance of Bacteroidetes has been shown to be substantially accentuated as a consequent exercise intervention in an early obesity and NAFLD model and in controls, compared with corresponding untrained group [100]. It should be pointed out that our LLS cohorts, mainly CENT, had a low PASE score, which indicates a sedentary lifestyle, consistent with a reduction in the Bacteroidetes phylum in these subjects compared to controls.

Taxa significantly associated with longevity were correlated with dietary, lifestyle and clinical variables in both CENT and NON. Most taxa associated with CENT were correlated with DMS score and bowel function. As for the former, most of the bacteria belonged to the Firmicutes phylum. Specifically, taxa belonging to the Clostridiaceae (*Clostridium*), Lachnospiraceae (Dorea), Peptostreptococcaceae (*Peptoniphilus*) and Thermicanaceae (*Thermicanus*) families positively correlated with DMS scores. The Clostridiaceae family has been associated with increased dietary fibers in rodent models [101], responding to dietary carbohydrates. Interestingly, fiber represents one of the nutrients with a beneficial impact evaluated for the attribution of adherence score to the Mediterranean diet [50]. Clostridium species can ferment carbohydrates, proteins, organic acids, and other organics, and produce acetic acid, propionic acid, butyric acid (SCFAs) and some solvents, such as acetone and butanol. SCFAs and most of the metabolites they produce, such as bile acids (BAs), proteins and other metabolic substances, bring many benefits to gut health [102]. As for *Dorea*, its ability to produce SCFAs from vegetables has been reported [103]. In CENT*, Catenibacterium* correlated negatively with DMS scores. This bacterial genus belongs to the Erysipelotrichidae family (Firmicutes), which has been associated with high fat diets in humans and in rodent models [104,105] and with inflammation-related intestinal disease and metabolic disorders [106], although a subsequent study observed an increase related to their diets [107] in a group of Egyptian adolescents compared to US Children.

In regard to bowel function, eight bacterial taxa were positively correlated, and one was negatively correlated. The positively correlated taxa mostly belonged to the Firmicutes phylum (Thermicanaceae, *Thermicanus, E. inopinata),* but also included Desulfohalobiaceae, *Desulfonauticus*, *D. autotrophicus* (Proteobacteria) and the Synergistetes phylum, with its Synergistaceae family. The taxa negatively correlated with bowel function belonged to the Firmicutes phylum (*L. taiwanensis*). Further studies are needed to clarify the significance of these correlations with the intestinal function of the study subjects, given that, to our knowledge, the literature does not describe the physiological implications of these taxa on human health. Diet may mediate some of these correlations, as Thermicanaceae and *Thermicanus* are positively associated with both DM and bowel function.

In CENT, *Desulfovibrio* negatively correlated with MMSE and MNA scores and to former alcohol consumption. *D. piger* negatively correlated with ADL, MNA and PASE scores. This data is not surprising, given the pro-inflammatory implications of these taxa (discussed above).

In NON there was a greater number of correlations with DMS, to the number of medications and to PASE score. Members belonging to the Actinobacteria phylum were all negatively correlated to DMS score (Bifidobacteriaceae, *Bifidobacterium*, *Bifidobacterium* spp., Streptomycetaceae), as well as those belonging to Bacteroidetes (*Bacteroides clarus*) and Firmicutes (*Lactobacillus ultunensis*), whereas others belonging to Bacteroidetes (Odoribacteraceae, *Bacteroides dorei*, *Pedobacter kwangyangensis* and *Parabacteroides* spp.), Firmicutes (*Alkaliphilus*, *Clostridium frigoris*, *Lactobacillus ultunensis*, *Peptoniphilus*) and Proteobacteria (*Bilophila* and *Bilophila wadsworthia*) were positively correlated. Taxa belonging to the Actinobacteria phylum, such as *Bifidobacterium*, *Bifidobacterium* spp., *Blautia wexlerae* (Firmicutes) and Bacteroidaceae (Bacteroidetes phylum) were negatively correlated with the number of medications, while the Bacteroidetes phylum (and related *Bacteroides*, *Bacteroides* spp., *Sphingobacterium shayense*) and Alcaligenaceae and *Sutterella* (Proteobacteria phylum) showed positive correlations. Several significant associations have also been found for other clinical variables, such as ADL score and PASE scores. Taxa belonging to the Proteobacteria phylum (*Serratia*, *S. entomophila*, Escherichia, *E. albertii*) and *Eggerthella* (Actinobacteria) were inversely related to ADL scores; however, in relation to PASE score, Firmicutes members (Thermicanaceae, *Thermicanus*, *Blautia wexlerae*) on one hand, and Bacteroidetes and Proteobacteria members on the other, showed positive and negative correlations, respectively.

We performed a comparative prediction analysis of the functional metagenome using PICRUSt.

It was previously observed that older adults and long-lived subjects have reduced pathways related to carbohydrate metabolism and amino acid synthesis [6]. It should be pointed out that, with aging, dietary habits change due to a reduction in appetite, loss of teeth, decrease in gustatory perception and decreased efficiency of the digestive system, which results in a reduction in the absorption of essential nutrients [6]. In this regard, we observed a depletion in glycan metabolism in both CENT and NON, and a reduction of starch and sucrose metabolism (carbohydrate degradation related pathway) in NON. This result agrees with a previous study carried out on a group of subjects recruited in the same territory (Sardinia, Italy), in which a reduction of pathways related to carbohydrate degradation was observed compared to elderly and younger subjects [43]. In our study cohort, this finding is of particular interest in the light of the contextual reduction of the Bacteroidetes phylum in both CENT and NON. In fact, Bacteroidetes encode more carbohydrate-degrading enzymes than Firmicutes (more representative than Bacteroidetes in our long-lived cohorts) and possess a lesser number of ABC carbohydrate transporters. Genes encoding ABC transporters specific for glycans are often located adjacent to those encoding glycoside hydrolases (with which they are co-expressed) in Firmicutes but not in Bacteroidetes; this might be a glycan acquisition strategy that Firmicutes have evolved [108]. Furthermore, ABC transporters are involved in the transport of a variety of substrates, including nutrients, toxins, antibiotics and xenobiotics [109]. NON showed a significant increase in ABC transporter expression, which may be related to more frequent use of medications in this cohort.

Another noteworthy finding of our research is the significant increase in the two-component system pathway in both CENT and NON compared to CTLs, which points out the greater adaptability of the long-lived intestinal microbial ecosystem compared to that of younger subjects. Two-component signal transduction systems represent a means of communication through which bacteria perceive and respond to their environment, including stress conditions, nutrient availability, quorum signals, chemokines, *p*H and other factors [110]. It is a strategy developed by bacteria to adapt their cellular physiology to changes in the environment. The importance of such a sophisticated signaling mechanism justifies their prevalence throughout the bacterial kingdom [111].

Assuming that the microbial phenotypic patterns observed in our cohort of centenarians were peculiar and/or predisposing to the state of longevity, we hypothesized that they were not significantly divergent in terms of abundance from those of the paired centenarian’s offspring cohort, considering both genetic and environmental effects as predisposing factors for the state of longevity. As expected, CPAR and COFF differed significantly only in five bacterial taxa, which lost statistical significance following Bonferroni’s post-hoc correction. In CPAR, we have observed a reduction in taxa belonging to the Firmicutes phylum, such as *Faecalibacterium*, *Faecalibacterium prausnitzii* (Clostridiaceae) and *Roseburia faecis* (Lachnospiraceae), as well as a reduction in *Bacteroides denticanum* and *Bacteroides plebeius*, which belong to the Bacteroidetes phylum. Furthermore, no statistically significant differences in alpha and beta diversity, nor in metabolic function, between CPAR and COFF were observed. These findings were not surprising and seemed to confirm our hypothesis. The genetic make-up and environmental factors, such as diet, geographical environment, type of residence, modality of childbirth or type of breastfeeding, act synergistically as a multifactorial cause in the modulation of GM. In fact, it should be noted that all subjects of our COFF cohort were born through natural childbirth and were nursed with their mother’s milk and all, except one, lived in their own home and maintained an identical diet to that of their parents until adulthood. Moreover, their diet remained similar until before the sampling of the study.

## 5. Conclusions

In conclusion, long-lived subjects were more similar to each other than to younger controls and the greatest divergences, in terms of microbial composition, emerged from the comparison between nonagenarians and controls. Nonagenarians showed an increase in both anti and pro-inflammatory bacterial taxa compared to younger subjects. This is not entirely surprising, given that these are subjects in an advanced aging phase, thus subjects in whom the likelihood of successful extreme aging, as we see in centenaries, is unknown. In this context, a prospective analysis of the nonagenarian population might be useful in order to understand what intestinal microbial pattern would predispose a subject to reach the age of a hundred.

Our population of centenarians diverged less from younger subjects in term of bacterial taxa compared to NON. Overall, the main biomarkers associated with centenarians belonged to the Verrucomicrobia phylum, including the *Akkermansia muciniphila* species, considered to be a significant biomarker of gut homeostasis for its ability to promote intestinal integrity; at the same time, there is a significant increase in taxa with an anti-inflammatory phenotype, biomarkers of a state of health. This intestinal microbial ecosystem could guarantee intestinal health, which would then translate into the health of the whole organism.

The results deriving from the GM analysis of centenarians’ offspring suggest that genetics and environmental factors act synergistically as a multifactorial cause in the modulation of GM towards a phenotype similar to that of centenarians, although these findings need to be confirmed by larger study cohorts and by prospective studies in order to clarify whether such microbial phenotypic patterns are predisposing factors in longevity.

## Figures and Tables

**Figure 1 nutrients-14-02436-f001:**
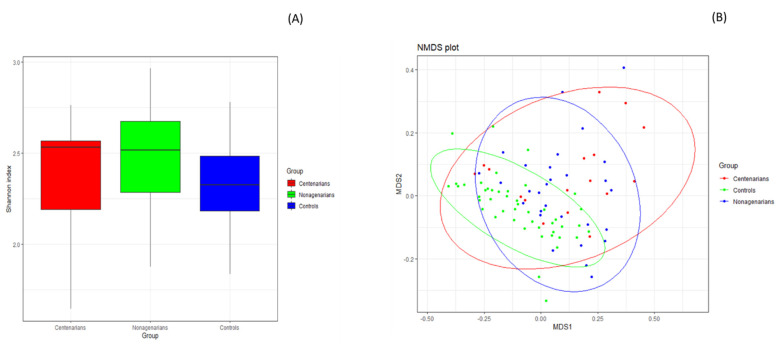
GM alpha and beta diversity analysis between CENT, NON and CTLs groups. (**A**) Plots indicate a statistically significant difference in the Shannon index between CENT and NON, evaluated by Kruskal-Wallis test. *p* equal to or less than 0.05 was considered statistically significant. (**B**) The Non-Metric Multidimensional Scaling (NMDS) plot based on Bray-Curtis distance matrix, performed in R software v.3.5.2 (ggplot2 package), showed a marked separation between the GM communities of LLS groups and CTLs. The statistical significance among the groups was determined with Permutational Multivariate Analysis of Variance (PERMANOVA) performed in R-vegan, function adonis (sum of square s = 1.498, mean of squares = 0.749, F = 6.074, R = 0.1201, *p* = 0.001). Significant segregation persisted only in the comparison between CENT and CTLs (*p* = 0.006) and between NON and CTLs (*p* = 0.003) following the pairwise PERMANOVA test performed in R (RVAdeMemoire package). *p* ≤ 0.05 was considered statistically significant. CENT = centenarian subjects, NON = nonagenarian subjects, CTLs = healthy younger controls, MDS = Multidimensional Scaling.

**Figure 2 nutrients-14-02436-f002:**
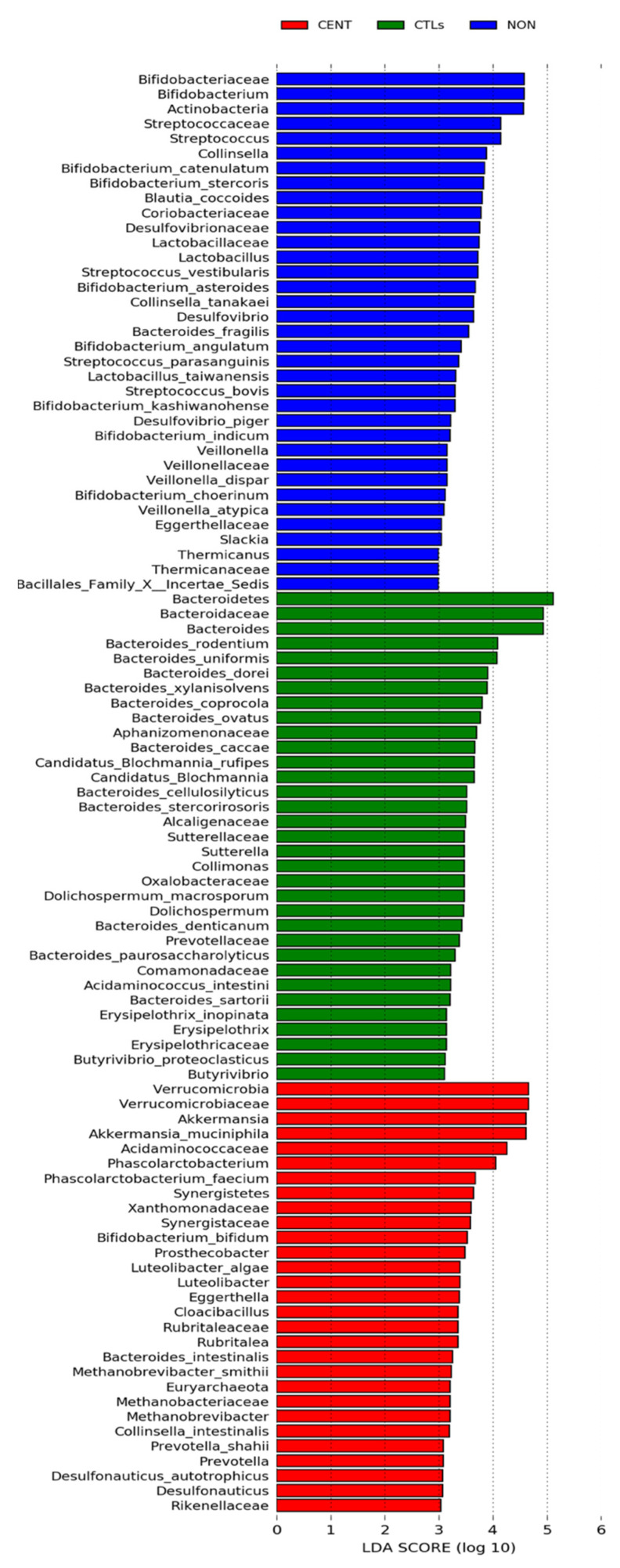
Linear Discriminant Analysis Effect Size (LEfSe) of microbial taxa between CENT, NON and CTLs.

**Figure 3 nutrients-14-02436-f003:**
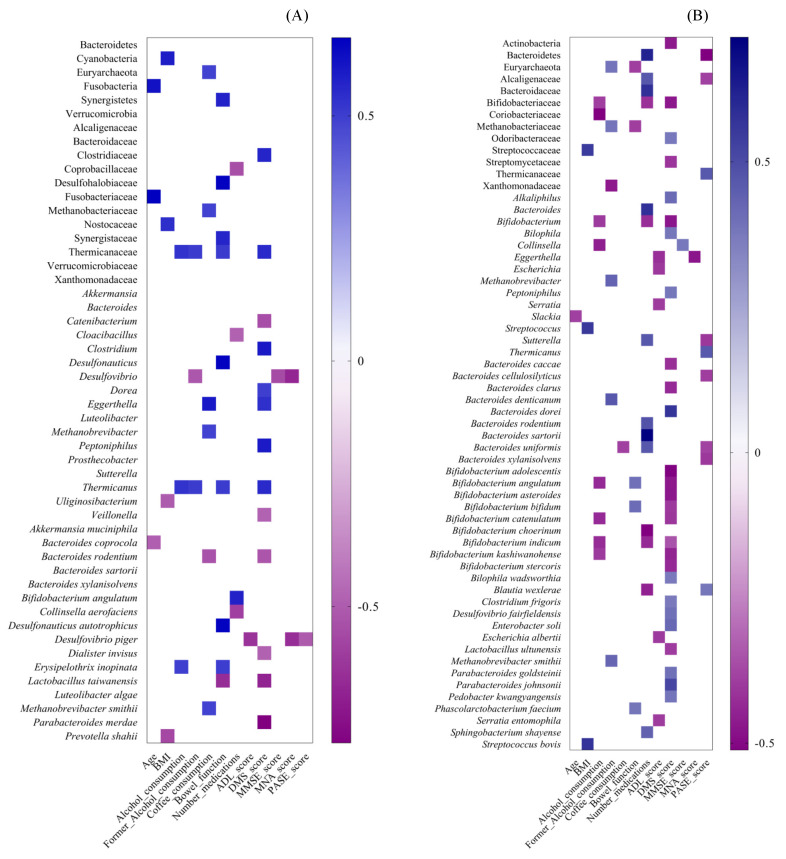
Spearman correlation analysis between GM alterations and clinical variables in CENT and NON. Heatmaps were generated in GraphPad Prism v.7.0d. A correlation heatmap was used to represent significant statistical correlation values (Rho) between intestinal microbiota taxa significantly associated with CENT (**A**), NON (**B**) and clinical features. In the heatmap, violet squares indicate significant negative correlations (Rho < 0.0, *p* ≤ 0.05) and blue squares indicate significant positive correlations (Rho > 0.0, *p* ≤ 0.05). Only *p* ≤ 0.05 are shown. BMI = Body Mass Index, ADL = Activities of Daily Living, DMS = Mediterranean Diet score, MMSE = Mini Mental State Evaluation, MNA = Mini Nutritional Assessment, PASE = Physical Activity Scale for the Elderly.

**Figure 4 nutrients-14-02436-f004:**
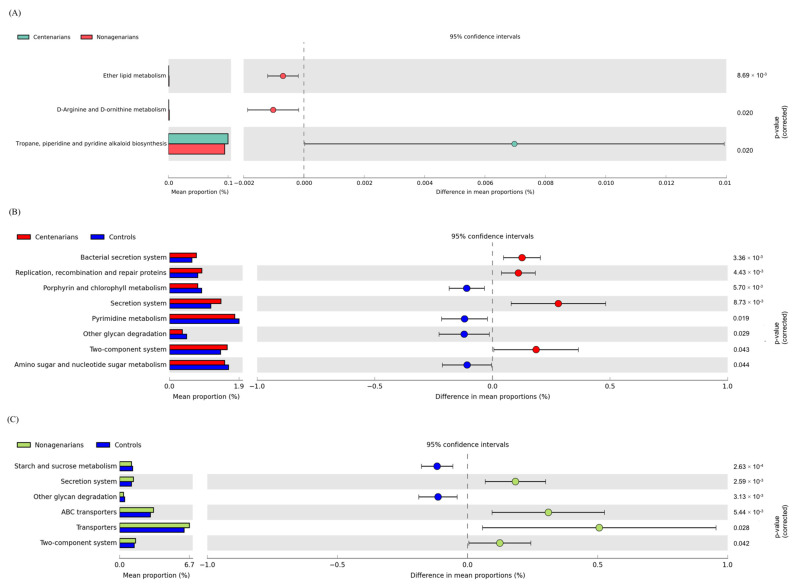
Statistically significant functional alterations of intestinal microbiome between CENT and NON and between LLS and CTLs. Phylogenetic Investigation of Communities by Reconstruction of Unobserved States (PICRUSt) algorithm was performed on Galaxy software v.1.0. (https://galaxy.morganlangille.com/), accessed on 18 May 2020, to infer metagenome composition in the samples by analyzing OTUs generated by QIIME pipeline. Bacterial metabolic pathways were predicted and classified by the Kyoto Encyclopedia of Genes and Genomes (KEGG). Statistical differences were analyzed for all metabolism pathways using the Statistical Analysis of Metagenomic Profiles (STAMP) software. Statistical significance was tested using Welch’s test, with a Storey False Discovery Rate correction (FDR) correction. *q* equal to or less than 0.05 was considered statistically significant. (**A**) Pathways more abundant in CENT are on the positive side (green circle with 95% CI); pathways less abundant in CENT are on the negative side (red circle with 95% CI). Mean proportions are shown in stacks (CENT = green; NON = red). The difference in mean proportions indicates the mean proportion CENT minus the mean proportion NON. (**B**,**C**) Pathways more abundant are on the positive side (red or green circle with 95% CI). Pathways less abundant are on the negative side (blue circle with 95% CI). Mean proportions are shown in stacks (CENT = red, NON = green; CTLs = blue. The difference in mean proportions indicates the mean proportion LLS group minus the mean proportion CTLs. CENT = centenarian subjects, NON = nonagenarian subjects, CTLs = healthy younger controls, LLS = long-lived subjects.

**Table 1 nutrients-14-02436-t001:** Clinical and lifestyle data of CENT and CTLs subjects.

	CENT	NON	CTLs	*p* CENT *vs* NON	*p* CENT *vs* CTLs	*p* NON *vs* CTLs	
*n*	17	29	46				
**Demographic data**							
Age (M ± SD)	102.2 ± 2.3	93.1 ± 2.7	50.9 ± 8.3	**1.62** × 10^−**14**^	**1.11** × 10^−**33**^	**3.46** × 10^−**39**^	
Female (*n*, %)	14, 82.3	23, 79.3	37, 80.4	0.802	0.863	0.906	
Male (*n*, %)	3, 17.6	6, 20.7	9, 19.6	0.802	0.863	0.906	
**Anthropometric data**							
BMI (M ± SD)	26.12 ± 4.5	27.14 ± 4.2	22.75, 2.8	0.274	**0.001**	**0**	
**Lifestyle factors**							
Current smoking status (*n*, %)	0, 0.0	0, 0.0	7, 15.2	-	0.088	**0.027**	
Former smoking status (*n*, %)	2, 11.8	3, 10.3	2, 4.3	0.881	0.785	0.881	
Current alcohol consumption (*n*, %)	8, 47.1	11, 37.9	9, 19.6	0.544	0.701	0.829	
Former alcohol consumption (*n*, %)	13, 76.5	17, 58.6	n.d.	0.22	-	-	
Coffee consumption (*n*, %)	13, 76.5	21, 72.4	33, 71.7	0.762	0.982	0.677	
Bowel function							
Movements/week (M ± SD)	4.4 ± 2.2	4.9 ± 2.1	n.d.	0.193	-	-	
From 1 to 3 (movements/week, %)	8, 47.1	10, 30.5	n.d.	n.d.	-	-	
From 4 to 5 (movements/week, %)	3, 17.6	2, 6.9	n.d.	n.d.	-	-	
From 6 to 7 (movements/week, %)	6, 35.3	17, 58.6	n.d.	n.d.	-	-	
Medications, n/day (M ± SD)	3.4 ± 2.9	4.5 ±2.8	n.d.	0.209	**4.43** × 10^−**09**^	**1.12** × 10^−**14**^	
MMSE score (M ± SD)	19.3 ± 4.3	25.3 ± 4.3	n.d.	**0.001**	-	-	
MDS (M ± SD)	31.0 ± 5.3	30.7 ± 4.5	32.9 ± 3.7	0.426	0.067	**0.036**	
ADL score (M ± SD)	2.7 ± 2.3	4.2 ± 2.3	n.d.	**0.03**	-	-	
PASE score (M ± SD)	11.2 ± 12.3	31.7 ± 24.1	n.d.	**0.024**	-	-	
MNA score (M ± SD)	24.1 ± 3.4	24.0 ± 6.9	n.d.	0.782	-	-	
Comorbidities (*n*, %)	16, 94.1	27, 93.1	6, 13.0	0.893	**0**	**0**	

M = mean, *n* = number, SD = standard deviation, BMI = Body Mass Index, MMSE = Mini Mental State Evaluation, MDS = Mediterranean Diet score, ADL = Activities of Daily Living, PASE = Physical Activity Scale for the Elderly, MNA = Mini Nutritional Assessment, n.d. = not determined. The statistical significance was evaluated by *t* tests for independent samples for continuous variables and by Pearson’s chi-squared test for categorical variables in SPSS software v.28.0.1.0. Bold values denote statistical significance (*p* ≤ 0.05). CENT = centenarians, NON = nonagenarians, CTLs = healthy younger controls.

**Table 2 nutrients-14-02436-t002:** Clinical and lifestyle data of CPAR and COFF subjects.

	CPAR	COFF	*p* CPAR *vs* COFF
*n*	7	7	
**Demographic data**			
Age (M ± SD)	102 ± 1.9	65.4 ± 6.6	**0.000**
Female (*n*, %)	5, 71.4	4, 57.1	1.000
**Anthropometric data**			
BMI (M ± SD)	27.76 ± 5.3	25.93 ± 1.9	0.425
**Lifestyle factors**			
Current smoking status (*n*, %)	0, 0.0	1, 14.3	1.000
Former smoking status (*n*, %)	5, 71.4	6, 85.7	0.500
Current alcohol consumption (*n*, %)	3, 42.9	6, 85.7	0.375
Former alcohol consumption (*n*, %)	5, 71.4	6, 85.7	1.000
Coffee consumption (*n*, %)	6, 85.7	7, 100	1.000
Bowel function			
Movements/week (M ± SD)	3.7 ± 1.9	6.4 ± 1.1	**0.037**
From 1 to 3 (movements/week, %)	4, 57.1	0, 0.0	n.d.
From 4 to 5 (movements/week, %)	2, 28.6	1, 14.3	n.d..
From 6 to 7 (movements/week, %)	1, 14.3	6, 85.7	n.d.
Medications, n/day (M ± SD)	5.0 ± 3.6	2.1 ± 1.7	**0.041**
MMSE score (M ± SD)	20.5 ± 4.3	27.2 ± 2.7	**0.014**
MDS (M ± SD)	28.6 ± 7.9	27.0 ± 7.0	0.323
ADL score (M ± SD)	2.4 ± 2.6	6.0 ± 0.0	**0.010**
PASE score (M ± SD)	10.5 ± 12.8	133.5 ± 22.3	**0.000**
MNA score (M ± SD)	24.0 ± 4.2	27.0 ± 2.5	**0.023**
Comorbidities (*n*, %)	7, 100	7, 100	1.000

M = mean, *n* = number, SD = standard deviation, BMI = Body Mass Index, MMSE = Mini Mental State Evaluation, MDS = Mediterranean Diet score, ADL = Activities of Daily Living, PASE = Physical Activity Scale for the Elderly, MNA = Mini Nutritional Assessment, n.d. = not determined. The statistical significance was evaluated by *t* test for independent samples for continuous variables and by Pearson’s chi-squared test for categorical variables in SPSS software v.28.0.1.0. Bold values denote statistical significance (*p* ≤ 0.05). CPAR = centenarian parents, COFF = centenarians’ offspring.

**Table 3 nutrients-14-02436-t003:** Firmicutes/Bacteroidetes ratio analysis between CENT, NON and CTLs and between CPAR and COFF.

	MEAN ± SD	OVERALL *P*	BONFERRONI *P* (CENT VS. CTLS)	BONFERRONI *P* (NON VS. CTLS)
		**0.003**	**0.015**	**0.022**
**CENT**	4.88 ± 4.79			
**NON**	3.83 ± 4.28			
**CTLS**	1.73 ± 1.85			
		0.499		
**CPAR**	4.88 ± 4.79			
**COFF**	3.83 ± 4.28			

The statistical significance was calculated by the non-parametric Mann-Whitney test in SPSS software v.28.0.1.0. Pairwise comparison was performed only for significant taxa, followed by Bonferroni correction for multiple comparisons. The Firmicutes/Bacteroidetes ratio was significantly higher in CENT and NON compared to CTLs (*p* = 0.015 and *p* = 0.022 respectively). No statistical significance persisted in the comparison between CPAR and COFF (*p* = 0.499). Bold values denote statistical significance (*p* ≤ 0.05). CENT = centenarian subjects, NON = nonagenarian subjects, CTLs = healthy younger controls, CPAR = centenarian parents, COFF = centenarians’ offspring.

**Table 4 nutrients-14-02436-t004:** Statistically significant differences in the relative abundance of bacterial taxa between CENT, NON and CTLs.

					Post hoc Analysis, Bonferroni Method (Only for Significant Bacteria)						
Phylum	Family	Genus	Species	Kruskal-Wallis *p*-Value	Pairwise Group	Pairwise *p*-Value	Chi Square (χ^2^)	↓/↑	Mean ± SD CENT	Mean ± SD NON	Mean ± SD CTLs
Actinobacteria				0.0018	CENT- CTLs	0.0393	12.66	↑	8.04 ± 10.23	9.12 ± 9.91	3.09 ± 3.61
					NON- CTLs	0.0039		↑			
Actinobacteria	Bifidobacteriaceae			0.0037	NON- CTLs	0.0058	11.2	↑	6.95 ± 10.25	8.01 ± 9.71	2.08 ± 3.03
Actinobacteria	Coriobacteriaceae			0.0003	CENT- CTLs	0.0445	16.16	↑	1.10 ± 1.13	1.17 ± 1.41	0.56 ± 0.87
					NON- CTLs	0.0004		↑			
Actinobacteria	Bifidobacteriaceae	*Bifidobacterium*		0.0039	NON- CTLs	0.0061	11.1	↑	6.91 ± 10.18	7.98 ± 9.67	2.07 ± 3.02
Actinobacteria	Coriobacteriaceae	*Collinsella*		0.0022	NON- CTLs	0	12.26	↑	0.66 ± 0.73	0.77 ± 1.28	0.00 ± 0.00
					CENT- CTLs	0		↑			
Actinobacteria	Coriobacteriaceae	Eggerthella		0.0075	CENT- CTLs	0.0275	9.79	↑	0.10 ± 0.22	0.05 ± 0.05	0.03 ± 0.05
					NON- CTLs	0.0394		↑			
Actinobacteria	Eggerthellaceae	*Slackia*		0.0024	NON- CTLs	0.0028	12.07	↑	0.27 ± 0.29	0.29 ± 0.26	0.17 ± 0.26
Actinobacteria	Bifidobacteriaceae	*Bifodobacterium*	*B. angulatum*	0	NON- CTLs	0	24.71	↑	0.01 ± 0.03	0.18 ± 0.60	0.00 ± 0.01
					CENT- NON	0.0035	24.71	↓			
Actinobacteria	Bifidobacteriaceae	*Bifodobacterium*	*B. asteroides*	0.0006	CENT- CTLs	0.0294	14.95	↑	0.11 ± 0.18	0.11 ± 0.14	0.02 ± 0.03
					NON- CTLs	0.0011		↑			
Actinobacteria	Bifidobacteriaceae	*Bifodobacterium*	*B. bifidum*	0.0175	NON- CTLs	0.0418	8.09	↑	0.29 ± 0.48	0.11 ± 0.17	0.04 ± 0.17
Actinobacteria	Bifidobacteriaceae	*Bifodobacterium*	*B. catenulatum*	0.0025	CENT- NON	0.0347	11.95	↓	0.92 ± 3.03	1.03 ± 2.04	0.12 ± 0.33
					NON- CTLs	0.0031	11.95	↑			
Actinobacteria	Bifidobacteriaceae	*Bifodobacterium*	*B. choerinum*	0.0118	NON- CTLs	0.033	8.88	↑	0.14 ± 0.19	0.24 ± 0.33	0.09 ± 0.19
Actinobacteria	Bifidobacteriaceae	*Bifodobacterium*	*B. indicum*	0.0018	NON- CTLs	0.0022	12.68	↑	0.14 ± 0.25	0.23 ± 0.25	0.06 ± 0.10
Actinobacteria	Bifidobacteriaceae	*Bifodobacterium*	*B. kashiwanohense*	0.0011	NON- CTLs	0.0008	13.55	↑	0.12 ± 0.34	0.16 ± 0.29	0.02 ± 0.05
Actinobacteria	Bifidobacteriaceae	*Bifodobacterium*	*B. stercoris*	0.04	NS	NS	6.44		1.30 ± 2.75	1.26 ± 2.20	0.27 ± 0.46
Actinobacteria	Coriobacteriaceae	*Collinsella*	*C. aerofaciens*	0.0122	NON- CTLs	0.0126	8.82	↑	0.36 ± 0.42	0.40 ± 0.48	0.21 ± 0.41
Actinobacteria	Coriobacteriaceae	*Collinsella*	*C. intestinalis*	0.0048	CENT- CTLs	0.0295	10.66	↑	0.13 ± 0.22	0.06 ± 0.06	0.09 ± 0.33
					NON- CTLs	0.0192		↓			
Actinobacteria	Coriobacteriaceae	*Collinsella*	*C. tanakaei*	0.001	NON- CTLs	0.0007	13.77	↑	0.09 ± 0.34	0.29 ± 1.26	0.01 ± 0.02
Bacteroidetes				0.0002	CENT- CTLs	0.0032	16.95	↓	24.74 ± 24.26	25.28 ± 17.60	43.59 ± 21.64
					NON- CTLs	0.0019		↓			
Bacteroidetes	Bacteroidaceae			0.0007	CENT- CTLs	0.0085	14.54	↓	15.77 ± 17.53	15.03 ± 12.52	28.05 ± 18.05
					NON- CTLs	0.004		↓			
Bacteroidetes	Rikenellaceae			0.0023	CENT- CTLs	0.0034	12.17	↑	0.12 ± 0.18	0.06 ± 0.11	0.07 ± 0.14
Bacteroidetes	Bacteroidaceae	*Bacteroides*		0.0007	CENT- CTLs	0.0085	14.54	↓	15.77 ± 17.53	15.03 ± 12.52	28.05 ± 18.05
					NON- CTLs	0.004		↓			
Bacteroidetes	Prevotellaceae	*Paraprevotella*		0.0495	NS	NS	NS		0.11 ± 0.20	0.18 ± 0.36	0.34 ± 0.51
Bacteroidetes	Porphyromonadaceae	*Porphyromonas*		0.1131	NS	NS	4.36		0.02 ± 0.02	0.19 ± 0.48	0.06 ± 0.09
Bacteroidetes	Bacteroidaceae	*Bacteroides*	*B. caccae*	0.0005	NON- CTLs	0.0003	15.31	↓	0.56 ± 0.76	0.27 ± 0.65	0.90 ± 1.09
Bacteroidetes	Bacteroidaceae	*Bacteroides*	*B. cellulosilyticus*	0.011	NON- CTLs	0.0091	9.02	↓	0.37 ± 0.74	0.19 ± 0.56	0.61 ± 1.22
Bacteroidetes	Bacteroidaceae	*Bacteroides*	*B. coprocola*	0.0209	CENT- CTLs	0.0333	7.73	↓	0.64 ± 2.45	0.28 ± 0.83	0.98 ± 3.70
Bacteroidetes	Bacteroidaceae	*Bacteroides*	*B. denticanum*	0.0154	NON- CTLs	0.0363	8.35	↓	0.13 ± 0.31	0.12 ± 0.23	0.52 ± 1.20
Bacteroidetes	Bacteroidaceae	*Bacteroides*	*B. dorei*	0.0338	NS	NS	6.78		1.05 ± 2.14	1.47 ± 3.18	2.21 ± 3.23
Bacteroidetes	Bacteroidaceae	*Bacteroides*	*B. fragilis*	0.0172	NON- CTLs	0.0316	8.13	↑	0.22 ± 0.30	0.68 ± 1.54	0.24 ± 0.71
Bacteroidetes	Bacteroidaceae	*Bacteroides*	*B. intestinalis*	0.0184	CENT- NON	0.0202	7.99	↑	0.22 ± 0.53	0.00 ± 0.01	0.11 ± 0.54
Bacteroidetes	Bacteroidaceae	*Bacteroides*	*B. ovatus*	0.0085	CENT- CTLs	0.0179	9.54	↓	0.27 ± 0.51	0.44 ± 0.63	1.27 ± 2.57
Bacteroidetes	Bacteroidaceae	*Bacteroides*	*B. paurosaccharolyticus*	0.0167	CENT- CTLs	0.0176	8.18	↓	0.10 ± 0.13	0.15 ± 0.18	0.18 ± 0.17
Bacteroidetes	Bacteroidaceae	*Bacteroides*	*B. rodentium*	0.0002	CENT- CTLs	0.0446	17.37	↓	1.82 ± 2.47	0.95 ± 1.11	2.65 ± 2.36
					NON- CTLs	0.0002		↓			
Bacteroidetes	Bacteroidaceae	*Bacteroides*	*B. sartorii*	0.0002	CENT- CTLs	0.0006	17.3	↓	0.22 ± 0.60	0.12 ± 0.10	0.25 ± 0.20
					NON- CTLs	0.0102		↓			
Bacteroidetes	Bacteroidaceae	*Bacteroides*	*B. stercorirosoris*	0	NON- CTLs	0	23.17	↓	0.40 ± 0.41	0.22 ± 0.24	0.62 ± 0.52
Bacteroidetes	Bacteroidaceae	*Bacteroides*	*B. uniformis*	0.0043	NON- CTLs	0.0042	10.89	↓	3.05 ± 5.92	1.43 ± 2.30	3.09 ± 3.36
Bacteroidetes	Bacteroidaceae	*Bacteroides*	*B. xylanisolvens*	0.0005	CENT- CTLs	0.0017	15.19	↓	0.50 ± 0.47	0.74 ± 0.82	1.74 ± 2.57
					NON- CTLs	0.014		↓			
Bacteroidetes	Prevotellaceae	*Paraprevotella*	*P. clara*	0.0378	CENT- CTLs	0.0453	6.55	↓	0.04 ± 0.08	0.06 ± 0.11	0.16 ± 0.28
*Bacteroidetes*	*Prevotellaceae*	*Prevotella*	*P. shahii*	0.0276	NS	NS	7.18		0.14 ± 0.59	0.02 ± 0.08	0.07 ± 0.21
Bacteroidetes	Sphingobacteriaceae	*Sphingobacterium*	*S. shayense*	0.0389	NON- CTLs	0.0421	4.49	↓	0.09 ± 0.10	0.08 ± 0.10	0.18 ± 0.27
Chloroflexi	Caldilineaceae			0.0288	NON- CTLs	0.0267	7.09	↓	0.06 ± 0.04	0.05 ± 0.06	0.12 ± 0.13
Cyanobacteria				0.0093	NON- CTLs	0.0071	9.36	↓	0.62 ± 0.81	0.31 ± 0.45	0.95 ± 1.63
*Cyanobacteria*	*Aphanizomenonaceae*	*Dolichospermum*		0.0019	CENT- CTLs	0.0045	12.57	↓	0.00 ± 0.00	0.01 ± 0.01	0.34 ± 1.30
					NON- CTLs	0.0348		↓			
Cyanobacteria	Aphanizomenonaceae	*Dolichospermum*	*D. macrosporum*	0	CENT- CTLs	0.0005	26.32	↓	0.00 ± 0.00	0.00 ± 0.00	0.34 ± 1.30
					NON- CTLs	0		↓			
Euryarchaeota				0	CENT- CTLs	0	19.43	↑	0.29 ± 0.65	0.12 ± 0.36	0.00 ± 0.00
					NON- CTLs	0.0131		↑			
Euryarchaeota	Methanobacteriaceae			0	CENT- CTLs	0	40.68	↑	0.29 ± 0.65	0.12 ± 0.36	0.03 ± 0.20
					NON- CTLs	0.013		↑			
Euryarchaeota	Methanobacteriaceae	*Methanobrevibacter*	*M. smithii*	0	CENT- CTLs	0	25.27	↑	0.28 ± 0.61	0.11 ± 0.34	0.04 ± 0.19
					NON- CTLs	0.0116		↑			
Euryarchaeota	Methanobacteriaceae	*Methanobrevibacter*		0	CENT- CTLs	0	24.45	↑	0.29 ± 0.65	0.12 ± 0.36	0.04 ± 0.20
					NON- CTLs	0.0189		↑			
Firmicutes	Eubacteriaceae			0.0179	NON- CTLs	0.0445	8.05	↑	0.13 ± 0.10	0.14 ± 0.12	0.09 ± 0.08
Firmicutes	Lactobacillaceae			0.009	CENT- NON	0.0267	9.43	↓	0.23 ± 0.47	1.04 ± 3.22	0.13 ± 0.17
					NON- CTLs	0.0235	9.43	↑			
Firmicutes	Streptococcaceae			0	NON- CTLs	0	23.55	↑	0.72 ± 0.87	1.80 ± 2.18	0.19 ± 0.26
Firmicutes	Synergistaceae			0	CENT- CTLs	0.0067	30.49	↑	0.48 ± 0.88	0.17 ± 0.63	0.03 ± 0.13
Firmicutes	Thermicanaceae			0.0051	CENT- CTLs	0.0244	10.55	↑	0.11 ± 0.21	0.14 ± 0.23	0.03 ± 0.06
					NON- CTLs	0.0251		↑			
Firmicutes	Acidaminococcaceae	*Acidaminococcus*		0.0084	NON- CTLs	0.006	9.56	↓	0.36 ± 1.22	0.04 ± 0.13	0.54 ± 1.64
Firmicutes	Lachnospiraceae	*Blautia*		0.0376	CENT- CTLs	0.0335	6.56	↓	3.63 ± 2.36	5.48 ± 3.94	6.50 ± 4.86
Firmicutes	Lachnospiraceae	*Butyrivibrio*		0.0016	CENT- CTLs	0.0038	12.93	↓	0.02 ± 0.02	0.02 ± 0.03	0.12 ± 0.34
					NON- CTLs	0.0318		↓			
Firmicutes	Syntrophomonadaceae	Caldicellulosiruptor		0.0102	CENT- CTLs	0.0148	9.17	↑	0.11 ± 0.11	0.08 ± 0.10	0.06 ± 0.07
Firmicutes	Eubacteriaceae	*Eubacterium*		0.0296	CENT- NON	0.0283	7.04	↓	0.13 ± 0.21	0.30 ± 0.82	0.05 ± 0.10
Firmicutes	Lactobacillaceae	*Lactobacillus*		0.0064	NON- CTLs	0.0174	10.12	↑	0.22 ± 0.46	1.00 ± 3.09	0.12 ± 0.16
					CENT- NON	0.0204	10.12	↓			
Firmicutes	Acidaminococcaceae	*Phascolarctobacterium*		0.0197	NON- CTLs	0.0393	7.85	↓	1.84 ± 2.79	0.29 ± 0.44	1.35 ± 2.00
Firmicutes	Streptococcaceae	*Streptococcus*		0	NON- CTLs	0	23.67	↑	0.70 ± 0.87	1.78 ± 2.16	0.19 ± 0.25
Firmicutes	Bacillales_Family X_Incertae Sedis	*Thermicanus*		0.0051	CENT- CTLs	0.0244	10.55	↑	0.11 ± 0.21	0.14 ± 0.23	0.03 ± 0.06
					NON- CTLs	0.0251		↑			
Firmicutes	Acidaminococcaceae	*Acidaminococcus*	*A. intestini*	0.001	NON- CTLs	0.0009	13.81	↓	0.02 ± 0.06	0.00 ± 0.00	0.14 ± 0.42
Firmicutes	Lachnospiraceae	*Blautia*	*B. coccoides*	0.0072	CENT- CTLs	0.0059	9.87	↓	0.63 ± 0.46	1.40 ± 1.31	1.40 ± 1.05
					CENT- NON	0.0375	9.87	↓			
Firmicutes	Lachnospiraceae	*Blautia*	*B. wexlerae*	0.0526	NS	NS	5.89		0.29 ± 0.42	0.59 ± 0.84	0.88 ± 1.84
*Firmicutes*	*Lachnospiraceae*	*Butyrivibrio*	*B. proteoclasticus*	0.0016	CENT- CTLs	0.0039	12.89	↓	0.02 ± 0.02	0.02 ± 0.03	0.12 ± 0.34
					NON- CTLs	0.0327		↓			
Firmicutes	Erysipelothricaceae	*Erysipelothrix*	*E. inopinata*	0.0317	CENT- CTLs	0.0258	6.9	↓	0.06 ± 0.13	0.13 ± 0.29	0.16 ± 0.49
Firmicutes	Lactobacillaceae	*Lactobacillus*	*L. taiwanensis*	0.0001	CENT- CTLs	0.0313	18.86	↑	0.02 ± 0.05	0.16 ± 0.84	0.00 ± 0.00
					NON- CTLs	0.0001		↑			
Firmicutes	Acidaminococcaceae	*Phascolarctobacterium*	*P. faecium*	0.0048	NON- CTLs	0.0252	10.67	↓	0.68±1.10	0.04±0.12	0.45±1.03
					CENT- NON	0.0095	10.67	↑			
Firmicutes	Ruminococcaceae	*Ruminococcus*	*R. torques*	0.0437	NS	NS	NS		0.27 ± 0.58	0.13 ± 0.26	0.14 ± 0.30
Firmicutes	Streptococcaceae	*Streptococcus*	*S. bovis*	0	NON- CTLs	0	20.22	↑	0.03 ± 0.03	0.24 ± 0.55	0.03 ± 0.13
Firmicutes	Streptococcaceae	*Streptococcus*	*S. parasanguinis*	0	NON- CTLs	0	24.27	↑	0.05 ± 0.08	0.19 ± 0.30	0.01 ± 0.01
Firmicutes	Streptococcaceae	*Streptococcus*	*S. vestibularis*	0.0066	NON- CTLs	0.0046	10.05	↑	0.18 ± 0.31	0.57 ± 0.89	0.05 ± 0.10
Firmicutes	Veillonellaceae	*Veillonella*	*V. atypica*	0.0122	NON- CTLs	0.0091	8.82	↑	0.05 ± 0.14	0.12 ± 0.22	0.04 ± 0.11
Firmicutes	Veillonellaceae	*Veillonella*	*V. dispar*	0.0195	NON- CTLs	0.0155	7.87	↑	0.05 ± 0.18	0.11 ± 0.28	0.02 ± 0.04
Fusobacteria	Fusobacteriaceae			0.0266	NS	NS	7.26		0.03 ± 0.10	0.10 ± 0.35	0.21 ± 1.39
Proteobacteria	Alcaligenaceae			0.0003	CENT- CTLs	0.0065	16.38	↓	0.46 ± 1.31	0.46 ± 1.43	0.83 ± 0.86
					NON- CTLs	0.0014		↓			
Proteobacteria	Comamonadaceae			0.0357	NS	NS	NS		0.07 ± 0.11	0.11 ± 0.33	0.13 ± 0.18
Proteobacteria	Desulfohalobiaceae			0.014	CENT- CTLs	0.0109	8.53	↑	0.23 ± 0.27	0.11 ± 0.09	0.15 ± 0.27
Proteobacteria	Xanthomonadaceae			0	CENT- CTLs	0.0015	45.43	↑	0.13 ± 0.15	0.09 ± 0.11	0.01 ± 0.03
					NON- CTLs	0.0021		↑			
Proteobacteria	Oxalobacteraceae	Collimonas		0	CENT- CTLs	0	61.68	↓	0.00 ± 0.00	0.00 ± 0.00	0.32 ± 0.54
					NON- CTLs	0		↓			
Proteobacteria	Desulfohalobiaceae	*Desulfonauticus*		0.0142	CENT- CTLs	0.011	8.51	↑	0.23 ± 0.27	0.11 ± 0.09	0.15 ± 0.27
Proteobacteria	Desulfovibrionaceae	*Desulfovibrio*		0.0079	CENT- CTLs	0.0177	9.68	↑	0.44 ± 0.48	0.70 ± 1.66	0.19 ± 0.34
Proteobacteria	Enterobacteriaceae	*Enterobacter*		0.0246	NON- CTLs	0.0201	7.41	↑	0.34 ± 0.67	0.76 ± 2.35	0.11 ± 0.35
Proteobacteria	Enterobacteriaceae	*Escherichia*		0.0022	NON- CTLs	0.0019	12.28	↑	7.00 ± 12.18	3.14 ± 6.70	0.22 ± 0.65
Proteobacteria	Yersiniaceae	*Serratia*		0.0007	NON- CTLs	0.0005	14.56	↑	1.15 ± 1.92	0.67 ± 1.10	0.06 ± 0.13
Proteobacteria	Sutterellaceae	*Sutterella*		0.0003	CENT- CTLs	0.0068	16.32	↓	0.43 ± 1.17	0.46 ± 1.43	0.80 ± 0.84
					NON- CTLs	0.0015		↓			
Proteobacteria	Zoogloeaceae	*Uliginosibacterium*		0.0327	NON- CTLs	0.0356	6.84	↓	0.02 ± 0.07	0.00 ± 0.00	0.20 ± 1.25
Proteobacteria	Enterobacteriaceae	*Candidatus Blochmannia*	*C. B. rufipes*	0.0118	NON- CTLs	0.0123	8.89	↓	0.00 ± 0.00	0.00 ± 0.00	0.65 ± 0.76
Proteobacteria	Desulfohalobiaceae	*Desulfonauticus*	*D. autotrophicus*	0.0142	CENT- CTLs	0.011	8.51	↑	0.23 ± 0.27	0.11 ± 0.09	0.15 ± 0.27
Proteobacteria	Desulfovibrionaceae	*Desulfovibrio*	*D. piger*	0.0004	CENT- CTLs	0.0093	15.42	↑	0.14 ± 0.36	0.20 ± 0.72	0.03 ± 0.09
					NON- CTLs	0.002		↑			
Proteobacteria	Enterobacteriaceae	*Escherichia*	*E. albertii*	0.004	NON- CTLs	0.0037	11.06	↑	5.67 ± 9.94	2.54 ± 5.38	0.20 ± 0.60
Proteobacteria	*Yersiniaceae*	*Serratia*	*S. entomophila*	0.0009	NON- CTLs	0.0007	14.12	↑	1.13 ± 1.90	0.66 ± 1.09	0.06 ± 0.13
Synergistetes				0.0007	CENT- CTLs	0.0024	14.54	↑	0.56 ± 0.95	0.28 ± 0.70	0.04 ± 0.07
					NON- CTLs	0.0154		↑			
Synergistetes	Synergistaceae	*Cloacibacillus*		0.0028	CENT- CTLs	0.0018	11.77	↑	0.27 ± 0.71	0.11 ± 0.57	0.02 ± 0.11
Synergistetes	Synergistaceae	*Synergistes*		0.0191	CENT- CTLs	0.0148	7.92	↑	0.15 ± 0.45	0.03 ± 0.09	0.01 ± 0.06
Verrucomicrobia				0.0032	CENT- CTLs	0.0036	11.47	↑	10.26 ± 14.88	6.46 ± 10.39	2.20 ± 4.71
Verrucomicrobia	Verrucomicrobiaceae			0.0047	CENT- CTLs	0.0054	10.72	↑	10.19 ± 14.84	6.43 ± 10.36	2.19 ± 4.70
Verrucomicrobia	Verrucomicrobiaceae	*Akkermansia*		0.0054	CENT- CTLs	0.0058	10.45	↑	9.02 ± 13.17	5.67 ± 9.17	1.91 ± 4.12
Verrucomicrobia	Verrucomicrobiaceae	*Luteolibacter*		0.001	CENT- CTLs	0.0012	13.78	↑	0.49 ± 0.70	0.31 ± 0.50	0.11 ± 0.24
Verrucomicrobia	Verrucomicrobiaceae	*Prosthecobacter*		0.0072	CENT- CTLs	0.0098	9.86	↑	0.16 ± 0.22	0.10 ± 0.15	0.04 ± 0.08
Verrucomicrobia	Rubritaleaceae	*Rubritalea*		0.0035	CENT- CTLs	0.0045	11.29	↑	0.38 ± 0.53	0.24 ± 0.39	0.09 ± 0.18
Verrucomicrobia	Verrucomicrobiaceae	*Akkermansia*	*A. muciniphila*	0.0054	CENT- CTLs	0.0058	10.45	↑	9.02 ± 13.16	5.67 ± 9.17	1.91 ± 4.12
Verrucomicrobia	Verrucomicrobiaceae	*Luteolibacter*	*L. algae*	0.001	CENT- CTLs	0.0012	13.78	↑	0.49 ± 0.70	0.31 ± 0.50	0.11 ± 0.24

Table shows the GM significant differences between CENT, NON and CTLs performed by Kruskal-Wallis test on R software v.3.5.2. Pairwise comparison was performed only for significant taxa, followed by Bonferroni correction for multiple comparisons. *p* equal to or less than 0.05 was considered statistically significant. CENT = centenarian subjects, NON = nonagenarian subjects, CTLs = healthy younger controls, ↓ = significantly reduced in the first term of the pairwise group, ↑ = significantly increased in the first term of the pairwise group.

**Table 5 nutrients-14-02436-t005:** Relative abundance differences of bacterial taxa between CPAR and COFF.

					Post-Hoc Analysis, Bonferroni Method (only for Significant Bacteria)			
Phylum	Family	Genus	Species	Kruskal-Wallis *p*-Value	Bonferroni *p*	↓/↑	Mean ± SD CPAR	Mean ± SD COFF
Bacteroidetes	Bacteroidaceae	*Bacteroides*	*B. denticanum*	0.028	1.37	**↓**	0.05 ± 0.03	0.97 ± 0.84
Bacteroidetes	Bacteroidaceae	*Bacteroides*	*B. plebeius*	0.043	1.98	**↓**	0.06 ± 0.14	2.03 ± 2.10
Firmicutes	Ruminococcaceae	*Faecalibacterium*		0.018	0.90	**↓**	7.52 ± 4.48	12.83 ± 8.04
Firmicutes	Ruminococcaceae	*Faecalibacterium*	*F. prausnitzii*	0.028	1.37	**↓**	1.81 ± 1.34	3.34 ± 2.63
Firmicutes	Lachnospiraceae	*Roseburia*	*R. faecis*	0.028	1.37	**↓**	0.32 ± 0.23	1.26 ± 0.90

Relative abundance differences of bacterial taxa between CPAR and COFF were performed by Kruskal-Wallis test on R software v.3.5.2 followed by Bonferroni correction for multiple comparisons. Bonferroni *p* equal to or less than 0.05 was considered statistically significant. CPAR = centenarian parents, COFF = centenarians’ offspring, ↓ = significantly reduced in CPAR.

## Data Availability

The study was deposited in the European Nucleotide Archive (ENA) (https://www.ebi.ac.uk/ena), accessed on 13 May 2022, under the study accession number PRJEB52843 (https://www.ebi.ac.uk/ena/browser/view/PRJEB52843, accessed on 13 May 2022).

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
