# Peer review of "Gut Microbiota Markers and Dietary Habits Associated with Extreme Longevity in Healthy Sardinian Centenarians"

_nutrients, 2022, doi:10.3390/nu14122436_

Round 1
Reviewer 1 Report
Venessa Palmas and associates in their report describe results obtained from the study aimed to investigate the gut microbiota in groups of long-lived Sardinians. Performed analyses showed that in subjects with long healthy life expectancy (centenarians and nonagenarians) the microbial composition of gut change in a such a way that an increase in pro-inflammatory bacterial species is accompanied with increases in anti-inflammatory ones. Such changes in gut microbiota increase the adaptive ability to changing environment.
In general, the experiments were well planned and executed. Overall, the studies are described clearly and although, the reported results are confirmatory to some extent, they also provide some information that increase our understanding on relation between kind of diet gut microbial composition and life expectancy. In general, the draw conclusions are somewhat speculative (as mentioned by authors) especially concerning the synergy between genetics and environmental factors in promoting long life expectancy and required study on larger group of subjects.
Author Response
Dear Referee,
Thank you for reviewing the manuscript. We would like to thank you for the flattering judgment on our manuscript.
Best regards
Aldo Manzin
Reviewer 2 Report
Manuscript ID: nutrients-1761110
-In this manuscript, the authors studied about “Gut microbiota markers associated with lifestyle and diet in extreme longevity: focus on Sardinian healthy centenarians”. This paper can be interesting. However, there are some concerns before its acceptance. These points are valid in this manuscript process.
-Comments:
This paper has the potential to be fascinating. However, there are certain doubts before it is approved. In this manuscript procedure, these points are valid.
-Comments:
Q1: The title needs to be revised; it is ineffective, too long, and seems like the review manuscript title.
Q2: The quality of Figure 3 is poor. It needs to be improved. Microbiomes are a mystery to me.
Q3: The author could change the italic format of the bacterium names.
Q4: author could reduce conclusion part as well. They wrote to many irrelevant sentences.
Q5: Significance of human gut microbiome has documented. Cite this research articles appropriately in manuscript. https://doi.org/10.3390/ijms22031160.
Author Response
Dear Referee,
thank you for you valuable revision. We have made the changes you indicated in the Review Report Form of Reviewer 2. Specifically:
Q1: The title needs to be revised; it is ineffective, too long, and seems like the review manuscript title.
We thank for the comment. We have entered the following title: “Gut microbiota markers and dietary habits associated with extreme longevity in healthy Sardinian centenarians”. We changed the title also in the re-submitted “Supplementary material” file.
Q2: The quality of Figure 3 is poor. It needs to be improved. Microbiomes are a mystery to me.
We apologize for the inconvenience. We have inserted the Figure 3 in high resolution.Q3: The author could change the italic format of the bacterium names.We have used the italic format exclusively for genus and bacterial species according to the taxonomic nomenclature. We changed some bacteria names that were not written in italics format by mistake.
Q4: author could reduce conclusion part as well. They wrote to many irrelevant sentences.
The authors thanks for the comment. We have deleted some irrelevant sentences.
Q5: Significance of human gut microbiome has documented. Cite this research articles appropriately in manuscript. https://doi.org/10.3390/ijms22031160.
We have cited the indicated article and changed the number of subsequent citations in the test and in the bibliography.